# In Vitro and In Vivo Activity of 14-*O*-[(4,6-Diamino-pyrimidine-2-yl) thioacetyl] Mutilin against Methicillin-Resistant *Staphylococcus aureus*

**DOI:** 10.3390/molecules26113277

**Published:** 2021-05-28

**Authors:** Yunxing Fu, Chunqing Leng, Yuan Fan, Xia Ma, Xianghui Li, Xuefei Wang, Zhenghuan Guo, Xiujun Wang, Ruofeng Shang

**Affiliations:** 1Zhengzhou Key Laboratory of Immunopharmacology of Effective Components of Chinese Veterinary Medicine, College of Veterinary Medicine, Henan University of Animal Husbandry and Economy, Zhengzhou 450046, China; fyx1261648623@163.com (Y.F.); maxia801010@126.com (X.M.); huiskys@163.com (X.L.); hnmywxf001@126.com (X.W.); guozhenhuan126@126.com (Z.G.); 2Henan Provincial Research Center for the Inheritance and Innovation of Chinese Veterinary Medicine Classic Prescriptions, College of Veterinary Medicine, Henan University of Animal Husbandry and Economy, Zhengzhou 450046, China; 3Department of Animal Production, Jinhua Polytechnic, Jinhua 321000, China; 20191009@jhc.edu.cn; 4Key Laboratory of Veterinary Pharmaceutical Development, Ministry of Agriculture and Rural Affairs, Lanzhou Institute of Husbandry and Pharmaceutical Sciences of CAAS, Lanzhou 730050, China; fanyuan666@163.com

**Keywords:** DPTM, methicillin-resistant *Staphylococcus aureus* (MRSA), antibacterial activity, murine skin wound model, MIC

## Abstract

*Staphylococcus aureus* (*S. aureus*) is a major human pathogen that requires new antibiotics with unique mechanism. A new pleuromutilin derivative, 14-O-[(4,6-Diamino-pyrimidine-2-yl) thioacetyl] mutilin (DPTM), has been synthesized and proved as a potent antibacterial agent using in vitro and in vivo assays. In the present study, DPTM was further in vitro evaluated against methicillin-resistant *Staphylococcus aureus* (MRSA) isolated from dairy farms and outperformed tiamulin fumarate, a pleuromutilin drug used for veterinary. Moreover, a murine skin wound model caused by MRSA infection was established, and the healing effect of DPTM was investigated. The results showed that DPTM could promote the healing of MRSA skin infection, reduce the bacterial burden of infected skin MRSA and decrease the secretion of IL-6 and TNF-α inflammatory cytokines in plasma. These results provided the basis for further in-depth drug targeted studies of DPTM as a novel antibacterial agent.

## 1. Introduction

*Staphylococcus aureus* (*S. aureus*) is a major human pathogen associated with increased morbidity, mortality, and excess hospital costs [1,2]. It also causes skin and soft tissue infections (SSTI), including impetigo, folliculitis, furuncles, and subcutaneous abscesses [3,4]. Methicillin-resistant *Staphylococcus aureus* (MRSA) has aroused a growing concern and became a significant public health threat. Every year there are approximately 80,000 invasive infections individuals in the United States, resulting in 11,000 deaths annually [3,5,6]. Furthermore, decreased susceptibility and even resistance to vancomycin, daptomycin, linezolid, and other antibiotics, have been reported in many parts of the world [7,8,9]. As such, there is pressing need to develop novel antibiotics with unique mechanism of action against this dreadful pathogen. 

Pleuromutilin (Figure 1) is a natural compound that was first discovered and isolated from cultures of two species of basidiomycetes, *Pleurotus mutilus* and *P. passeckerianus* in 1951 [10]. Pleuromutilin derivatives selectively inhibited bacterial protein synthesis through interaction with prokaryotic ribosomes at the acceptor and donor site [11,12,13]. Modification of the glycolic ester side chain in pleuromutilin has been shown to give derivatives with improved antibacterial activities, and has led to tiamulin, valnemulin, retapamulin, and lefamulin (Figure 1) [14,15,16,17]. 

14-O-[(4,6-Diamino-pyrimidine-2-yl) thioacetyl] mutilin (DPTM, Figure 1) is a new pleuromutilin derivative with a pyrimidine moiety. It was first synthesized and has been shown excellent antibacterial activity, suggesting its potential as a promising antimicrobial drug [18,19]. In this study, we further investigated the activity of DPTM against MRSA using in vitro and in vivo assays.

## 2. Results 

### 2.1. Effect of DPTM In Vitro

The minimum inhibitory concentrations (MICs) of DPTM against two standard quality control strains of MRSA (ATCC 29213 and ATCC 33591) and a *S. aureus* (CMCC 26003) have been reported in our previous study [18]. Extensive panels of clinical isolates of MRSA (*n* = 54) were assessed for their susceptibility to DPTM. We chose tiamulin fumarate as reference drug because it was used primarily in veterinary medicine [20]. MICs for DPTM ranged from 0.0313 to 0.25 μg/mL, while that of tiamulin fumarate were 0.125 to 1 μg/mL (a full listing of this MIC data is in Appendix A). Within these collections of clinical isolates, MICs of DPTM were lower than that of tiamulin fumarate (Figure 2). 

### 2.2. Macroscopic Evaluation of Efficacy in a Murine Skin Wound Model

Bearing the excellent protection efficacy of mice infected with MRSA-29213 [18], DPTM was further assessed for its healing effect on murine skin wound caused by MRSA infections. After inoculation, symptoms of the skin wound and mouse viability were monitored daily for 6 days. Mice treated with three dosages of DPTM and retapamulin ointments had improved survival (100%) compared to positive control mice (80%). Mice that were not infected showed no death and normal wound with a small amount of thin exudate (Figure 3A). Most murine wounds in positive control group presented symptoms including the increased exudate, worsened with bloody skin, and heavy or purulent drainage (Figure 3B). Treatment with retapamulin and 1% and 2% DPTM caused improved symptoms with some exudate remaining, and the wounds begun healing at the margin with some crust (Figure 3C−E). However, wounds in 3% DPTM ointment treatment group exhibited partial thickening and the crust had peeled away from the margin leaving behind fresh skin (Figure 3F). 

### 2.3. Bacterial Count in Treated Skin Wound

Next, we test the efficacy of a topical antibiotic treatment in the skin wound model in which mice infected with MRSA were treated with three dosages of DPTM ointment (Figure 4). The bacterial counts in 2% DPTM, 3% DPTM, and retapamulin ointments treated mice were 4.40, 5.42, and 4.82 logs lower, respectively, (*p* < 0.01) than that in positive control mice, whereas the 1% DPTM ointment treated group was 1.02 log lower. The 3% DPTM ointment treated group had the bacterial count lowered by 1.05 log compared to 2% DPTM ointment treated group (*p* < 0.01), but only lowered by 0.63 log compared to retapamulin ointment treated group. 

### 2.4. White Blood Cell (WBC) Level

To further investigate the efficacy of DPTM on the murine skin wound infected with MRSA, we analyzed the WBC which are involved in protecting the body against both infectious disease and foreign invaders [21] in blood, using a blood biochemistry analyzer (Hefei Jianneng Optical Instrument Co., Ltd., Hefei, China). The numbers of WBC in blank control group, three treatment groups and retapamulin ointment group decreased significantly compared with that in positive control group (*p* < 0.05). There was more WBC in the 1% DPTM ointment group compared with that in the retapamulin ointment group (*p* < 0.05). However, no significant difference was found among the 2% DPTM, 3% DPTM, and retapamulin ointment groups (Figure 5).

### 2.5. Histopathological Observation

The healing tissues obtained from all the six groups of animals in our excision wound model were processed for histopathological analysis by H&E and Masson staining (Figure 6). At 6 d post-infection, severe inflammatory cell infiltration, inflammatory cell accumulation, and tissue necrosis, as well as swelling and granular degeneration of muscle fiber were commonly observed in the wound skin of mice at the positive control group at the sixth day (Figure 6B_1_). It was worth noting that proliferation of myofibroblast, increase of new capillary, and attenuation of inflammatory reaction were noticeable after treatment with 2% and 3% DPTM and retapamulin ointments (Figure 6D_1_,E_1_,F_1_). Furthermore, obvious blue collagen deposition appeared in the wound after treatment with three dosages of DPTM and retapamulin ointments (Figure 6C_2_,D_2_,E_2_,F_2_). The connection of collagen fiber (dyed blue in Figure 6A_2_–F_2_) in treatments groups was closer than that in control group, especially the connection of collagen fiber in 3% DPTM ointment treatment group which showed significantly higher tightness than that of retapamulin ointment treatment group (Figure 6E_2_,F_2_).

### 2.6. IL-6, TNF-α, and VEGF Levels

Primary skin infections stimulate inflammatory response which plays an essential role in the defense against pathogens [22]. This response involves a complex interplay of cytokines and chemokines, such as the pro-inflammatory cytokines interleukin-6 (IL-6), tumor necrosis factor-α (TNF-α), and vascular endothelial growth factor (VEGF) [23]. Therefore, we used the ELISA method to detect the changes of IL-6, TNF-α, and VEGF after drug treatment. As illustrated in Figure 7, all treated groups, except 1% DPTM ointment, effectively decreased IL-6 and TNF-α induced by the inflammation in comparison with that in the control group (Figure 7A,B). To our surprise, no significant difference of VEGF secreted in serum between treatment groups and control group was observed (Figure 7C).

## 3. Discussion

At present, MRSA displays resistance to most of β-lactam antibiotics, including oxacillin, methicillin, amoxicillin, and penicillin [24,25]. Drug resistance acquired by MRSA has increased clinical risk and caused serious problems worldwide [26]. Therefore, it is of great significance to develop new drugs for the diseases infected by MRSA.

This study evaluated the potent antibacterial activities of DPTM, a derivative of pleuromutilin, against MRSA isolated from dairy farms and against MRSA ATCC43300 in a murine skin wound model. While DPTM showed excellent in vitro inhibition against standard strains of *S. aureu**s* and MRSA at previous MIC testing [18], activity against MRSA isolated from clinic is lacking. We found that DPTM inhibited 61.1% strains with the concentration of 0.0625 μg/mL, while no strain was inhibited with the same concentration of tiamulin fumarate.

*S. aureus* is the top infectious pathogens responsible for SSTIs in children and adults [27]. Retapamulin, a semi-synthetic member of pleuromutilin, has been licensed in USA and Europe as 1% ointment (Altabax) for the topical treatment of SSTIs caused by MRSA and *Streptococcus pyogenes* [28]. Therefore, we established the mouse MRSA ATCC43300 skin infection model to compare the therapeutic effect of DPTM with that of retapamulin. After treatment, DPTM reduced the exudation of the infected skin and accelerate the healing. Furthermore, 2% and 3% DPTM ointment significantly reduced the number of white blood cells in the blood, indicating that DPTM displayed a certain therapeutic effect on MRSA skin infections. The ability of drugs to promote healing of infectious wounds is related to its antibacterial activities, reducing inflammatory reactions, and promoting epithelial formation [22]. Therefore, the number of bacteria in the infected skin was counted to evaluate the antibacterial effect of DPTM in live animals. The results showed that 2% and 3% DPTM ointment significantly reduced the number of MRSA.

Inflammation is one of key stages for the healing of skin wounds proceeds. *S. aureus* activate the STAT3, MAPK, and NF-κB signaling pathways, which promote the expression and secretion of pro-inflammatory cytokines, such as IL-6 and TNF-α, in keratinocytes [29,30]. It is helpful for the recovery of skin wounds to inhibit the secretion of excessive pro-inflammatory cytokines. After treatment, DPTM ointment significantly reduced serum IL-6 and TNF-αsecretion, which indicates that DPTM could improve the inflammatory response caused by MRSA. As a chemoattractant, VEGF recruits macrophages and granulocytes and participates in nitric oxide-mediated vasodilation, which induces endothelial cells to participate wound healing, thereby promoting blood vessel formation and vascular remodeling [31,32]. However, in this study there is no significant difference of secreted VEGF in the serum between the control group and the treatment groups. Because we did not detect the concentration of VEGF during the initial time of administration, we could not conclude whether VEGF had reached the peak of expression, or DPTM could not affect the secretion of VEGF. This needs to be further studied in detail.

## 4. Materials and Methods

### 4.1. Reagents

The synthesis method of DPTM was described previously [18] in our lab. The purity of this compound was checked by Waters 2695 HPLC (Massachusetts, USA) and quantitative NMR analyses at 98.72%, and its structure was confirmed by IR, NMR (Appendix A), and HR-MS spectrometry. Tiamulin fumarate (purity: 98.5%) was purchased from Labor Dr. Ehrenstorfer-Schäfers (Augsburg, Germany) and retapamulin (98.0%) was purchased from (BOC Sciences, New York, NY, USA).

### 4.2. Bacterial Strain

All MRSA (n = 54) were isolated using brain-heart infusion (BHI) broth from fresh milk samples which were collected from different dairy farms in northwest China at 2017–2018. The isolates were identified by sequencing the 16S rRNA universal primer and Vitek 2 Compact (BioMerieux, Lyon, France), followed by typing the Staphylococcal chromosomal cassette mec (SCCmec) gene (Appendix A). Susceptibility testing was performed as per Clinical Laboratory and Standards Institute (CLSI) recommendations. MRSA ATCC43300 was purchased from Beijing Beina Science & Technology Co., Ltd. (Beijing, China).

### 4.3. Animals

Sixty healthy BALB/c mice (weight of 23 to 25 g; Centre of Experimental Animals of Lanzhou University, Lanzhou, China) were housed in a comfortable room and were given free access to standard diet and water. Mice were maintained on a 12 h light/dark cycle at the temperature of 25 °C and relative humidity of 55–65%. All animals were handled in strict accordance with good animal practice according to the Animal Ethics Procedures and Guidelines of the People’s Republic of China, and the study was approved by The Animal Administration and Ethics Committee of Lanzhou Institute of Husbandry and Pharmaceutical Sciences of CAAS (No. SYXK-2018-002).

### 4.4. MIC Determination

The MICs of DPTM were determined by micro-dilution technique in Mueller-Hinton broth (Beijing Solarbio Science & Technology Co., Ltd., Beijing, China) according to the Clinical and Laboratory Standards Institute (CLSI) guidelines [33]. The experiments were performed in triplicate.

### 4.5. Skin Infection Model

MRSA ATCC43300 was grown in tryptone soy broth (TSB, Beijing Solarbio Science & Technology Co., ltd. Beijing, China) at 37 °C overnight and harvested by centrifugation at 3000 g for 10 min, followed by being washed twice in phosphate buffer saline (PBS). Bacteria were suspended in sterile PBS at a concentration of 10^12^ CFU/mL. Different amounts of DPTM (0.05, 0.10, and 0.15 g) and retapamulin (0.10 g) used in this study were made as ointments (1%, 2%, and 3% DPTM and 2% retapamulin, respectively) with matrix including albolene (3 g), liquid paraffin (1.5 mL), lanolin (0.1 g), and anhydrous alcohol (0.4 mL). Wound preparation and infection protocol were modified from published report [34,35]. In brief, mice were randomly divided into six groups and were anaesthetized intraperitoneally with 10% chloral hydrate. The fur on mice back was shaved and the shaved area was cleaned with gauze sponges and water. An incision deep to sarcolemma was made through the shaved area after the alcohol disinfection. The blood from the wound was cleaned with cotton ball and an inoculum of 0.1 mL of MRSA (at a final concentration of approximately 10^12^ CFU/mL which was obtained by pre-test) in PBS was smeared evenly on the whole wound of mice except for the blank control group which was only treated with blank ointment matrix. After inoculation with MRSA for 4 h, the positive control group was treated with blank ointment matrix. The treatment groups were treated with 2% retapamulin and 1%, 2%, and 3% DPTM ointment, respectively. For each treatment 15 mg per mice of ointment was applied. The dosage for each group was continued for 5 days with 24 h intervals. After inoculation, the mice were caged separately and observed twice daily for their clinical signs and mortality. For all groups, the experiments were terminated 24 h (on day 6 after infection) after the last topical treatment in order to avoid carryover effects in vitro.

### 4.6. Detection of WBC, IL-6, TNF-α and VEGF in Blood Serum

On day 6 after infection, the blood was taken, by excising their eyeballs, for counting white blood cells and detecting the IL-6, TNF-α, and VEGF in blood serum by enzyme-linked immunosorbent assay (ELISA) using the commercially available kit (IL-16: Mouse IL-16 ELISA Kit, Boster Biological Technology co.ltd., Wuhan, China; TNF-α: Mouse TNF-αELISA Kit, Elabscience Biotechnology Co.,Ltd., Wuhan, China; VEGF: Mouse VEGF-B ELISA Kit, Elabscience Biotechnology Co.,Ltd., Wuhan, China).

### 4.7. Counting the Colonies of MRSA and Histological Examination

To examine CFU burden within the wound skin, the survival of the mice at 6 d after infection was used as the end-point and anaesthetized. The infected skins of five mice in treatment groups and positive control (including one dead mouse after infection) were sterilizing collected, homogenized, diluted (10×) and plated onto Baird-Parker agar (Hope Bio-Technology Co., Ltd., Qingdao, China) to count the colonies of MRSA after incubation for 16–24 h at 37 °C. The infected skins of the remaining five mice (including the other one dead mouse after infection) used for hematoxylin and eosin (H&E) and Masson trichrome staining after soaking in 10% formalin, respectively [19], and their lesions were observed using a Nikon DS-Fi2 fluorescent microscope (Nikon, Tokyo, Japan).

### 4.8. Statistical Analysis

Statistical analysis was performed using IBM SPSS Statistics for Windows version 24.0 (SPSS Inc., Chicago, IL, USA). The data were analyzed by One-way analysis of variance (ANOVA), followed by Dunnett’s post-hoc tests as appropriate. Statistically significant difference was defined as a *p* < 0.05 and the extremely significant difference was defined as a *p* < 0.01.

## 5. Conclusions

DPTM demonstrated potent in vitro activity against clinical isolates of MRSA with lower MICs than that of tiamulin fumarate. In in vivo efficacy using a murine skin wound model, 2% and 3% DPTM ointment displayed similar effect to retapamulin to significantly promote the healing of wound caused by MRSA and reduce bacterial count. In addition, DPTM decrease relative number of WBC and the secretion of IL-6 and TNF-α inflammatory cytokines in plasma. Thus, DPTM represents a promising treatment option for MRSA infections.

## Figures and Tables

**Figure 1 molecules-26-03277-f001:**
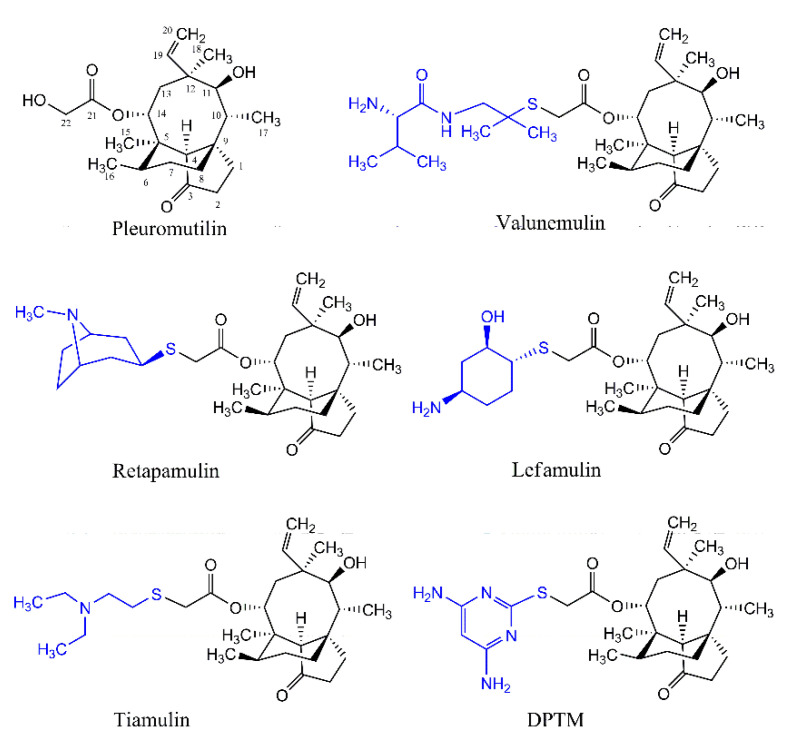
Chemical structures of pleuromutilin tiamulin, valnemulin, retapamulin, Lefamulin, and DPTM.

**Figure 2 molecules-26-03277-f002:**
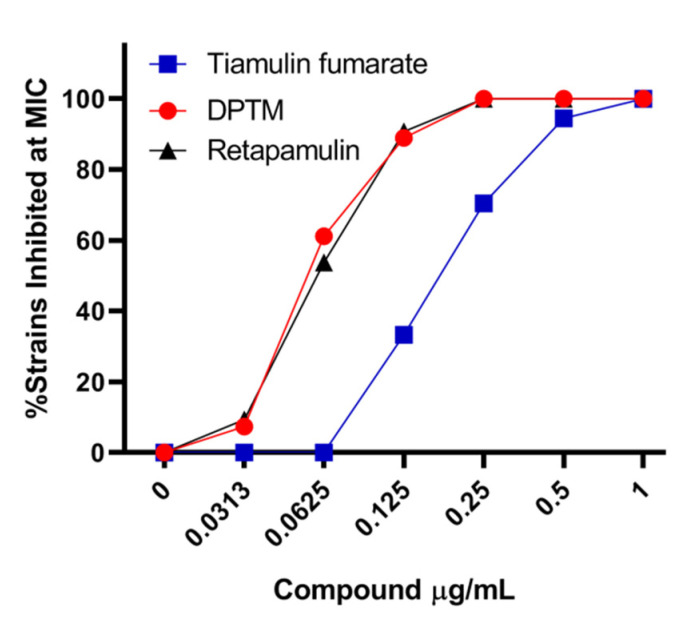
Assessment of DPTM and tiamulin fumarate against clinical isolates of MRSA. Full listing of this MIC data is in Appendix A.

**Figure 3 molecules-26-03277-f003:**
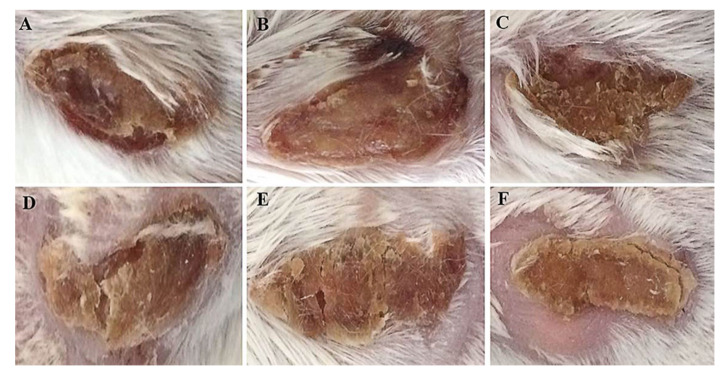
Macroscopic evaluation of wound at the 6th day in the infection model. Mice (n = 10 per group) were infected in the excision wound with MRSA ATCC43300 strain. (**A**) uninfected group, (**B**) control group, (**C**) 1% DPTM ointment treatment group, (**D**) 2% DPTM ointment treatment group, (**E**) 2% Retapamulin ointment treatment group, and (**F**) 3% DPTM ointment treatment group.

**Figure 4 molecules-26-03277-f004:**
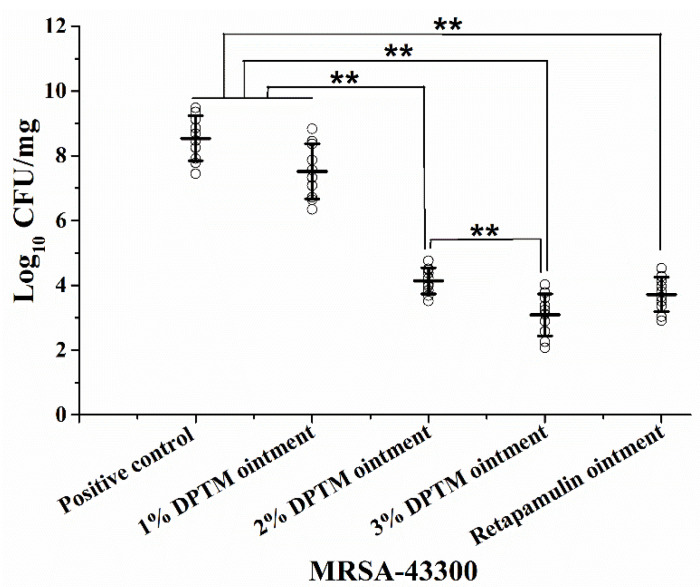
Efficacy of DPTM (1%, 2%, and 3% ointment) and retapamulin (2% ointment) against an experimental surgical wound infection in mice (n = 5) caused by MRSA. Values are means and standard errors of the means. ** *p* < 0.01.

**Figure 5 molecules-26-03277-f005:**
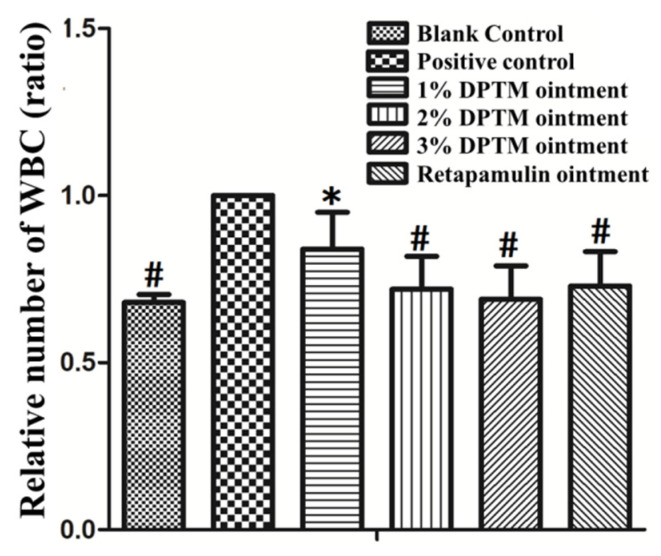
Effect of DPTM (1%, 2%, and 3% ointment) and retapamulin (2% ointment) on mice WBC. All values are mean ± SD. ^#^
*p* < 0.05 indicates that the blank group, 2% DPTM, 3% DPTM, and retapamulin ointment treatment group vs. control group; * *p* < 0.05 indicates that 1% DPTM ointment treatment group vs. 2% retapamulin group.

**Figure 6 molecules-26-03277-f006:**
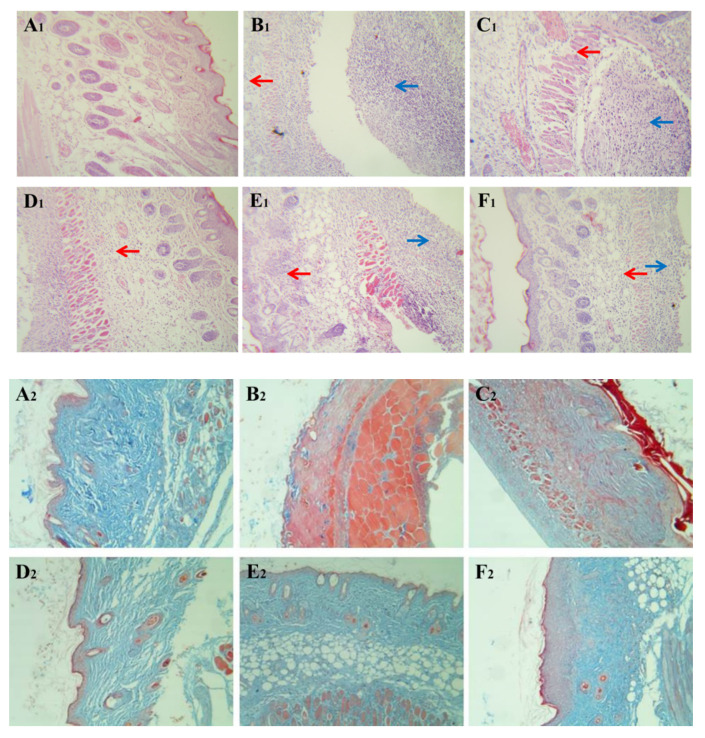
Effect of DPTM on pathological changes in MRSA infected skin (HE: 200×, Masson: 100×). These microscopic photographs of the wound healing skins were obtained from mice in the blank control group (**A_1_** and **A_2_**), positive control group (**B_1_** and **B_2_**), 1% DPTM ointment treatment group (**C_1_** and **C_2_**), 2% DPTM ointment treatment group (**D_1_** and **D_2_**), 2% retapamulin ointment treatment group (**E_1_** and **E_2_**), and 3% DPTM ointment treatment group (**F_1_** and **F_2_**). Red arrows point at inflammatory cell infiltration and the blue arrows point at inflammatory cell accumulation.

**Figure 7 molecules-26-03277-f007:**
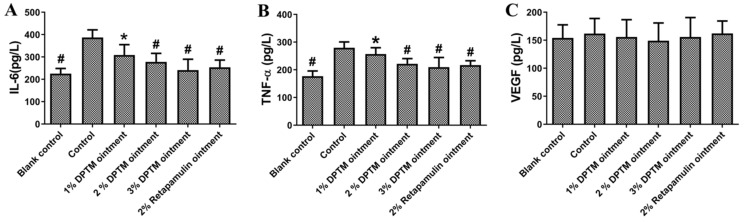
Effect of DPTM and retapamulin ointments on secretions of IL-6, TNF-α, and VEGF. IL-6 (**A**), TNF-α (**B**), and VEGF (**C**). All values are mean ± SD. ^#^
*p* < 0.05 indicates that the blank group, 2% DPTM, 3% DPTM, and 2% retapamulin ointment treatment group vs. the control group; * *p* < 0.05 indicates 1% DPTM ointment treatment group vs. the 2% retapamulin ointment group.

## Data Availability

The original contributions presented in the study are included in the article/Appendix A, further inquiries can be directed to the corresponding author.

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
