# Peer review of "In Vitro and In Vivo Activity of 14-O-[(4,6-Diamino-pyrimidine-2-yl) thioacetyl] Mutilin against Methicillin-Resistant Staphylococcus aureus"

_molecules, 2021, doi:10.3390/molecules26113277_

Round 1

Reviewer 1 Report

Previously, the authors reported the synthesis of DPTM and its antibacterial effect on two strains of S. aureus. The authors also demonstrated the in vivo efficacy of the compound in the murine intraperitoneal model. In this study, the authors expanded the panel of S. aureus to 54 MRSA strains for antibacterial effect and showed that the MICs of DPTM is generally lower than those of tiamulin fumarate, a reference compound. In a murine wound infection model, DPTM showed similar efficacy to that of Retapamulin, a pleuromutilin derivative in clinical usage, in wound healing and bacterial CFU counts.  As with retapamulin treatment, the treatment with DPTM lowered the white blood cell counts in the blood, reduced the tissue damage by the staphylococcal infection, and decreased the secretion of proinflammatory cytokines IL-6 and TNF-a .  Based on these results, the authors concluded that DPTM is a promising novel antibacterial agent.

Overall, the authors’ conclusions are supported by the presented data. Regarding the scientific content, I have a few concerns and recommendations. This manuscript also contains several writing errors, some of which are listed below.

Major comments

  1. For MIC comparison, the authors used Tiamulin fumarate, whereas, for wound infection, they used Retapamulin as a reference compound. Since Retapamulin is currently in clinical usage, it would be a better reference in MIC tests.
  2. 3A: without infection, the wound looks very severe. Why?
  3. Figure 6 and Page 5. Under “2. Histopathological Observation, “…. It was worth noting that … showed significantly higher tightness than that of retapamulin ointment treatment group (Figure 6E2 and F2)

: In Figure 6, it is difficult to see the histopathological changes described in the text. Provide pointers in the figure so that readers can see the histological changes the authors described in the text.

  1. Figure 7C and Page 6: Why no significant difference in VEGF is surprising?
  2. Discussion, Page 6. At present, MRSA displays resistance to all b-lactam antibiotics,

: This is not correct.  MRSA is sensitive to the 5th generation cephalosporin,  ceftaroline.

  1. Animal experiment: I do not see an animal protocol number approved by the authors’ institution.

Minor comments

Page 1, Introduction: sub cutaneous -> subcutaneous

Figures have no number.

Page 2: MICs of DPTM were more than that of … -> MICs of DPTM were lower than that of….

Page 3 and throughout the manuscript: Most murine wounds in positive control group..

             : In this manuscript, the authors call the no-drug treatment group a positive control. I think “negative control” would be more appropriate.

                : Also, instead of the” normal” group, the “uninfected” group might be better.

Fig. 4 legend: Values are means standard errors .. -> Values are means and standard errors..

Page 4. White Blood Cell (WBC) Lever -> White Blood Cell (WBC) Level

Page 4: For further investigate the efficacy of DPTM .. -> To further investigate the efficacy of DPTM

Page 6: To our surprised -> To our surprise

Page 7: DPTM ointment t significantly..  -> DPTM ointment significantly.. (delete “t”)

Page 7: DPTM ointment significantly reduced serum IL-6 and IL-1b secretion,

: No results were provided for IL-1b secretion. I guess the authors meant TNF-a.

Page 8, Under “4. Counting the Colonies of MRSA and Histological Examination”

                one died mouse -> one dead mouse

                after inoculation for 16-24 h -> after incubation for 16-24 h

                the other one died mouse -> the other dead mouse

Page 8, Under “Conclusion”

                We investigated the bacterial activities .. -> We investigated the antibacterial activities..

Author Response

  Previously, the authors reported the synthesis of DPTM and its antibacterial effect on two strains of S. aureus. The authors also demonstrated the in vivo efficacy of the compound in the murine intraperitoneal model. In this study, the authors expanded the panel of S. aureus to 54 MRSA strains for antibacterial effect and showed that the MICs of DPTM is generally lower than those of tiamulin fumarate, a reference compound. In a murine wound infection model, DPTM showed similar efficacy to that of retapamulin, a pleuromutilin derivative in clinical usage, in wound healing and bacterial CFU counts. As with retapamulin treatment, the treatment with DPTM lowered the white blood cell counts in the blood, reduced the tissue damage by the staphylococcal infection, and decreased the secretion of proinflammatory cytokines IL-6 and TNF-a. Based on these results, the authors concluded that DPTM is a promising novel antibacterial agent.

  Overall, the authors’ conclusions are supported by the presented data. Regarding the scientific content, I have a few concerns and recommendations. This manuscript also contains several writing errors, some of which are listed below.

Major comments

  1. For MIC comparison, the authors used tiamulin fumarate, whereas, for wound infection, they used retapamulin as a reference compound. Since retapamulin is currently in clinical usage, it would be a better reference in MIC tests.

     Author’s answers: I agree with the reviewer comment. In our previous studies (Yi et al., Eur. J. Med. Chem. 2017, 126, 687), tiamulin was used as a reference drug to compare with DPTM in the MIC test against the standard strains. In the MIC test of this study, we still used tiamulin as a reference drug to compare with DPTM against the clinical isolates. In the skin infection model, DPTM should be prepared as ointment and then could be used to test it anti-MRSA activity. Retapamulin was approved by the FDA as an ointment for skin infections like SSTIs caused by S. aureus. Furthermore, retapamulin and DPTM are all semisynthetic derivative of pleuromutilin. Therefore, we chose retapamulin as a reference drug to compare with DPTM in the skin infection model. Some information about tiamulin and retapamulin were already discussed in the “Discussion” part.

  1. 3A: without infection, the wound looks very severe. Why?

     Author’s answers: I agree with the reviewer comment. The wound in Figure 3A really looks very severe. However without infection, the wound displayed in Figure 3A did not show ulceration, exudate heavy or purulent drainage which are typical clinical symptoms caused by bacteria infection. But these symptoms could be observed in Figure 3B in which the wound was infected by MRSA. 

  1. Figure 6 and Page 5. Under “2. Histopathological Observation”, “…. It was worth noting that … showed significantly higher tightness than that of retapamulin ointment treatment group (Figure 6E2 and F2)”. In Figure 6, it is difficult to see the histopathological changes described in the text. Provide pointers in the figure so that readers can see the histological changes the authors described in the text.

     Author’s answers: Thank you for your recommend, Arrows have been added in the Figure 6B1-F1 to point the histological changes. In Figure A2-F2, the dyed blue part was the connection of collagen fiber which was needed not to point with arrow in Figure but was descripted in the manuscript.

  1. Figure 7C and Page 6: Why no significant difference in VEGF is surprising?

     Author’s answers: I think that VEGF could promote blood vessel formation and vascular remodeling. There are may be some changes of the secretion of VEGF during the wound healing. So it is surprising that no significant difference in VEGF in our experiment. However, I added some possible reasons for this result in “Discussion” part.  

  1. Discussion, Page 6. At present, MRSA displays resistance to all β-lactam antibiotics. This is not correct. MRSA is sensitive to the 5thgeneration cephalosporin, ceftaroline.

     Author’s answers: I really appreciate your careful checking. I have revised this error as “At present, MRSA displays resistance to most β-lactam antibiotics,”

  1. Animal experiment: I do not see an animal protocol number approved by the authors’ institution.

     Author’s answers: I have added the formal name of Ethics Committee and protocol number of our institution in Page 7 and in Page 9.

 Minor comments

  1. Page 1, Introduction: sub cutaneous -> subcutaneous

     Author’s answers: I have revised this clerical error.

  1. Figures have no number.

     Author’s answers: I have added the numbers for all the figures.

  1. Page 2: MICs of DPTM were more than that of … -> MICs of DPTM were lower than that of….

      Author’s answers: I thank you for your carefully check and the error has been revised.

  1. Page 3 and throughout the manuscript: Most murine wounds in positive control group. : In this manuscript, the authors call the no-drug treatment group a positive control. I think “negative control” would be more appropriate. Also, instead of the” normal” group, the “uninfected” group might be better.

        Author’s answers: Thank you for your recommendation. I think the positive control refers to “show” or “have” some symptom in clinical or physiological examination. While negative control refers to “not appear” or “not show” some symptom. However, I am very thankful for your recommendations to revise the” normal” group to the “uninfected” group. Now, this poor description has been revised.

  1. Fig. 4 legend: Values are means standard errors . -> Values are means and standard errors.

     Author’s answers: I have revised this error.

  1. Page 4. White Blood Cell (WBC) Lever -> White Blood Cell (WBC) Level

     Author’s answers: I appreciate your carefully check and the error has been revised.

  1. Page 4: For further investigate the efficacy of DPTM .. -> To further investigate the efficacy of DPTM.

     Author’s answers: I have revised this error.

  1. Page 6: To our surprised -> To our surprise.

     Author’s answers: I have revised this error.

  1. Page 7: DPTM ointment t significantly..  -> DPTM ointment significantly.. (delete “t”).

     Author’s answers: I have deleted “t” in this sentence.

  1. Page 7: DPTM ointment significantly reduced serum IL-6 and IL-1b secretion, : No results were provided for IL-1b secretion. I guess the authors meant TNF-a.

     Author’s answers: I appreciate your carefully check and the error has been revised.

  1. Page 8, Under “4. Counting the Colonies of MRSA and Histological Examination”

                one died mouse -> one dead mouse

                after inoculation for 16-24 h -> after incubation for 16-24 h

                the other one died mouse -> the other dead mouse

     Author’s answers: I have revised these errors.

  1. Page 8, Under “Conclusion”. We investigated the bacterial activities .. -> We investigated the antibacterial activities.

     Author’s answers: I have revised this error.

Reviewer 2 Report

The manuscript is both interesting and timely.  However there is one major concern regarding the bacterial strains used.  There is no description of the isolated strains.  How were these MRSA isolated and characterized?  How do they differ?  How was it determined that all 54 strains are actually different from one another?

The remainder of the study is well designed and the data is convincing.

Author Response

  The manuscript is both interesting and timely. However there is one major concern regarding the bacterial strains used. There is no description of the isolated strains. How were these MRSA isolated and characterized? How do they differ? How was it determined that all 54 strains are actually different from one another?

  The remainder of the study is well designed and the data is convincing.

  Author’s answers: I am very thankful for the reviewer’s question. We have added the isolation, identification and typing methods of the isolated strains simply. Each strain was isolated from different dairy firms and further typed by SCCmec gene. However, we did not sequence all the genes of bacteria to further actually differ them from one another.

Reviewer 3 Report

The present manuscript evaluated the antimicrobial effects of a new pleuromutilin derivative, 14-O-[(4,6-Diaminopyrimidine-2-yl) thioacetyl] mutilin (DPTM), over several S. aureus methicillin-resistant. Besides,  in a murine skin wound model caused by MRSA infection, the authors investigated the healing effect of DPTMed. As a result, the new molecule has lower MIC values when compared to a commercial antifungal and has a similar effect of a standard antifungal on the animal study.

The study was well-performed and improved the knowledge on the field. Nevertheless, I have some concerns about it, as follow:

1-The authors should present the Ethical committee of animal use approval.

2-The authors should justify why they choose only these three cytokines. Why did not the authors also analyze the IL-1beta? In the discussion section, the authors mentioned that IL-beta was analyzed; however, no results were shown.

3-Why different antifungals controls? In the MICs, the authors used tiamulin fumarate and in the animal study, used retapamulin ointments. It would be interesting to see the results of MIC of the retapamulin ointments and see the action of tiamulin fumarate on the animals.

4-Please remove the following from the conclusion: “We investigated the bacterial activities of DPTM against MRSA in vitro and in vivo” This is the objective of the study, not the conclusion.

5-Also, in the conclusion section, please inform that the results were similar to the standard antifungal used.

Author Response

The present manuscript evaluated the antimicrobial effects of a new pleuromutilin derivative, 14-O-[(4,6-Diaminopyrimidine-2-yl) thioacetyl] mutilin (DPTM), over several S. aureus methicillin-resistant. Besides, in a murine skin wound model caused by MRSA infection, the authors investigated the healing effect of DPTM. As a result, the new molecule has lower MIC values when compared to a commercial antifungal and has a similar effect of a standard antifungal on the animal study.

The study was well-performed and improved the knowledge on the field. Nevertheless, I have some concerns about it, as follow:

  1. The authors should present the Ethical committee of animal use approval.

Author’s answers: I have added the formal name of Ethics Committee and protocol number of our institution in Page 7 and in Page 9.

  1. The authors should justify why they choose only these three cytokines. Why did not the authors also analyze the IL-1beta? In the discussion section, the authors mentioned that IL-beta was analyzed; however, no results were shown.

  Author’s answers: I appreciate you for your carefully check. The IL-beta actually is TNF-a. Now this error has been revised.

  1. Why different antifungals controls? In the MICs, the authors used tiamulin fumarate and in the animal study, used retapamulin ointments. It would be interesting to see the results of MIC of the retapamulin ointments and see the action of tiamulin fumarate on the animals.

 Author’s answers: I agree with the reviewer comment. In our previous studies (Yi et al., Eur. J. Med. Chem. 2017, 126, 687), tiamulin was used as a reference drug to compare with DPTM in the MIC test against the standard strains. In the MIC test of this study, we still used tiamulin as a reference drug to compare with DPTM against the clinical isolates. In the skin infection model, DPTM should be prepared as ointment and then could be used to test it anti-MRSA activity. Retapamulin was approved by the FDA as an ointment for skin infections like SSTIs caused by S. aureus. Furthermore, retapamulin and DPTM are all semisynthetic derivative of pleuromutilin. Therefore, we chose retapamulin as a reference drug to compare with DPTM in the skin infection model. Some information about tiamulin and retapamulin were already discussed in the “Discussion” part.

  1. Please remove the following from the conclusion: “We investigated the bacterial activities of DPTM against MRSA in vitro and in vivo” This is the objective of the study, not the conclusion.

Author’s answers: I agree with your comment. This sentence was removed from conclusion.

  1. Also, in the conclusion section, please inform that the results were similar to the standard antifungal used.

Author’s answers: I am very thankful for your recommendations and I have informed that the results were similar to the standard antifungal used in the conclusion.

Round 2

Reviewer 2 Report

Much improved.  Genetic characterization of the different strains would be ideal, however it is beyond the scope of the current study

Author Response

Much improved. Genetic characterization of the different strains would be ideal. However it is beyond the scope of the current study.

Author’s answers: I agree with you for your comment. We have characterization of the different strains by SCCmec gene type. However, as you mentioned, this paper was aim to study the antibacterial activity of DPTM, not the isolation and molecular characteristics of strains.

Reviewer 3 Report

The authors addressed most of my previous concerns, except:

1-The authors should justify why they choose only these three cytokines. I suggest including the analysis of IL-1beta. 

2-  It would be interesting to see the results of MIC of the retapamulin

Author Response

The authors addressed most of my previous concerns, except:

  1. The authors should justify why they choose only these three cytokines. I suggest including the analysis of IL-1beta.

Author’s answers: Thank you for your suggestion. We chose these three cytokines because: VEGF contribute to endothelial cells to participate wound healing; TNF-α, IL-1β and IL-6 are all pro-inflammatory cytokines which contribute to verifying inflammatory changes after skin infections. In this study, we are regret that we only chose TNF-α and IL-6 as representative of many pro-inflammatory cytokines but not IL-1β. However, we could not add the analysis of IL-1β in the manuscript based on the current study because we did not detect the IL-1β in this study. If the review insists that the IL-1β should be analyzed, we have to re-study the efficacy of DPTM in a new murine skin wound model.

  1. It would be interesting to see the results of MIC of the retapamulin.

Author’s answers: Thank you for your suggestion. We have performed the study of MIC of the retapamulin against all the strains which were investigated on DPTM and tiamulin. Furthermore, the results were added in the manuscript and supplemental data.